# Molecular Characterization of Feline Parvovirus from Domestic Cats in Henan Province, China from 2020 to 2022

**DOI:** 10.3390/vetsci11070292

**Published:** 2024-06-30

**Authors:** Zuhua Yu, Wenjie Wang, Chuan Yu, Lei He, Ke Ding, Ke Shang, Songbiao Chen

**Affiliations:** 1Laboratory of Functional Microbiology and Animal Health, College of Animal Science and Technology, Henan University of Science and Technology, Luoyang 471003, China; 9902818@haust.edu.cn (Z.Y.);; 2Luoyang Key Laboratory of Live Carrier Biomaterial and Animal Disease Prevention and Control, Luoyang 471003, China; 3The Key Laboratory of Animal Disease and Public Health, Henan University of Science and Technology, Luoyang 471023, China; 4Pet & Human Health Engineering Technology Center, Luoyang Polytechnic, Luoyang 471900, China

**Keywords:** feline parvovirus, epidemiology, molecular characterization, domestic cats

## Abstract

**Simple Summary:**

In this study, 82 fecal samples of suspected FPV infection were collected from different areas of Henan Province from 2020 to 2022 (small sample in 2023), viral DNA was extracted, and FPV identification primers were used to identify 25 FPV-positive cases. VP2 and NS1 primers were further used to amplify the above positive samples, and the whole gene sequences of 11 VP2 and 21 NS1 strains were obtained and analyzed. Homology analysis showed that the amino acid homology of the VP2/NS1 gene of the isolates was 96.1–100% and 97.6–100%, respectively, with that of domestic and foreign endemic strains. The phylogenetic tree results showed that VP2 and NS1 of the local strains were mainly concentrated in the G1 subgroup, while the vaccine strains were distributed in the G3 subgroup, and the two strains were far-related. F81 cells were inoculated with local endemic strain SNF-01 (FPV-LY strain for short) from the FPV G1 subgroup for virus amplification, purification, and titer determination, and the pathogenesis of SNF-01 was detected. After five generations of blind transmission of F81 cells, the cells of the FPV-LY strain were rounded, wire-drawn, and crumpled. After sucrose density gradient centrifugation, the virus titer was determined by the Reed–Muench method to be 1.5 × 10^6^ TCID_50_/mL. Animal regression tests showed that the strain PFV-LY was highly pathogenic, and the cats showed typical clinical symptoms and pathological changes, and eventually died.

**Abstract:**

*Carnivore protoparvovirus*-1, *feline parvovirus* (FPV), and *canine parvovirus* (CPV) continue to spread in companion animals all over the world. As a result, FPV and CPV underwent host-to-host transfer in carnivorous wild-animal hosts. Here, a total of 82 fecal samples of suspected cat FPV infections were collected from Henan Province from 2020 to 2022. The previously published full-length sequence primers of VP2 and NS1 genes were used to amplify the targeted genes of these samples, and the complete gene sequences of 11 VP2 and 21 NS1 samples were obtained and analyzed. Analysis showed that the amino acid homology of the VP2 and NS1 genes of these isolates was 96.1–100% and 97.6–100%, respectively. The phylogenetic results showed that the VP2 and NS1 genes of the local isolates were mainly concentrated in the G1 subgroup, while the vaccine strains were distributed in the G3 subgroup. Finally, F81 cells were inoculated with the local endemic isolate Luoyang-01 (FPV-LY strain for short) for virus amplification, purification, and titer determination, and the pathogenesis of FPV-LY was detected. After five generations of blind transmission in F81 cells, cells infected with FPV-LY displayed characteristic morphological changes, including a round, threadlike, and wrinkled appearance, indicative of viral infection. The virus titer associated with this cytopathic effect (CPE) was measured at 1.5 × 10^6^ TCID_50_/mL. Subsequent animal regression tests confirmed that the virus titer of the PFV-LY isolate remained at 1.5 × 10^6^ TCID_50_/mL, indicating its highly pathogenic nature. Cats exposed to the virus exhibited typical clinical symptoms and pathological changes, ultimately succumbing to the infection. These results suggest that the gene mutation rate of FPV is increasing, resulting in a complex pattern of gene evolution in terms of host preference, geographical selection, and novel genetic variants. The data also indicate that continuous molecular epidemiological surveillance is required to understand the genetic diversity of FPV isolates.

## 1. Introduction

*Feline parvovirus* (FPV) is a single-stranded DNA virus and a variant of *Carnivore protoparvovirus* 1, belonging to the genus *Protoparvovirus* within the family *Parvoviridae*. A range of serious conditions including lethal disease, vomiting, enteritis, diarrhea, and acute lymphopenia in young animals are closely associated with *Carnivore protoparvovirus* 1 [1]. The main clinical manifestations of FPV are vomiting, gastroenteritis, and a high fever, resulting in huge economic losses to the global pet industry [2,3]. Here, FPV is the main causative agent of feline panleukopenia, which can also be caused by certain *canine parvovirus* (CPV) variants, CPV-2a, 2b, and 2c [4]. In 1998, Horiuchi et al. [5]. explored the evolutionary relationship between CPV and FPV, proposing that FPV is the ancestor of CPV; their phylogenetic analysis revealed that while the non-structural protein 1 (NS1) and capsid protein 2 (VP2) genes of FPV exhibited temporal changes, the VP2 protein itself remained largely unchanged. In contrast, CPV’s VP2 protein underwent alterations corresponding to genetic variations in its gene, suggesting a different evolutionary mechanism between the two viruses. FPV primarily evolved through random genetic drift, whereas CPV evolution occurred under the pressure of selection. Afterward, Parrish [6]. conducted a study on CPV evolution, pinpointing the emergence of CPV-2 and CPV-2a mutant strains in the late 1970s and early 1980s. Notably, CPV-2b strains, which replaced CPV-2a, became prevalent after 1986, demonstrating a significant evolutionary shift. Interestingly, the capsid protein gene sequence of an FPV isolated in 1990 closely resembled that of CPV-2b strains found in dogs. Further investigations across Europe, the United States, and Japan revealed the global dominance of the newer CPV-2a and CPV-2b variants, which replaced the original CPV-2 strain. This evolutionary transition coincided with CPV’s acquisition of a new host—the domestic cat. While CPV-2 strains were incapable of infecting cats, both CPV-2a and CPV-2b could replicate in felines, leading to approximately 10% of cat parvovirus cases being attributed to these newer variants. This shift underscores the dynamic interplay between viral evolution and host specificity, with CPV’s genetic adaptations enabling its cross-species transmission to cats [7]. Thus, members of the *Carnivore protoparvovirus* 1 group, which includes *mink enteritis virus* (MEV) and *raccoon parvovirus* (RaPV), could be host range variants [8]. For example, the FPV strain recently isolated from pandas demonstrated the ability to infect human cells [9], so there is a potential risk of transmission between felines and humans. Also, in general, the FPV and CPV vaccines are not effective in many wild canids and felids, resulting in a wild reservoir of the virus as well as creating conservation problems in endangered cats [10].

*Feline panleukopenia* is a common infectious disease of young animals in clinical practice, and has a high incidence and high treatment cost [11]. Because FPV is highly infectious and survives easily in the natural environment, cats at all stages can be infected. Severe diarrhea, hematochezia, leukopenia, and intestinal villi shedding are common clinical symptoms of FPV-infected cats. As a result, FPV poses a serious threat to the health of pet cats and wild cats [12]. Various strains of FPV have been identified through studies, with some potentially exhibiting higher virulence or increased resistance to existing vaccines. This underscores the critical need for ongoing research and surveillance to monitor the evolutionary dynamics of FPV [13]. While many studies have focused on analyzing the genetic evolution of FPV, particularly regarding the main capsid protein (VP2), limited information is available regarding the nonstructural gene NS1. Therefore, there exists a gap in understanding the role and evolution of NS1 in FPV, highlighting the importance of further investigation into this aspect of the virus’s genetics. Such research efforts can provide valuable insights into the mechanisms driving FPV evolution and aid in the development of more effective prevention and control strategies against this infectious disease. At present, FPV prevention and control is mainly based on multi-vaccine immunization. Since vaccines cannot fully effectively prevent and control viral infection, and the evolution of FPV may hinder molecular diagnosis, resulting in the virus not being detected and ultimately reducing the effectiveness of vaccines, a dilemma has arisen [14,15,16]. Therefore, it is important to understand the circulating and endemic strains of FPV and their variation, to analyze the characteristics of the genes controlling infectivity, such as VP2 and NS1, and to develop a novel candidate vaccine with good safety and immunogenicity that can prevent FPV infection.

In this study, fecal samples of domestic cats (*Felis silvestris catus*) suspected of FPV infection from different areas in Henan Province were collected from 2020 to 2022, and the viral VP2 and NS1 genes of the PCR-positive samples were sequenced and analyzed. This was aimed at clarifying the molecular genetic characteristics of circulating and endemic FPV strains in Henan Province, and studying their pathogenic characteristics. The results from this work will serve to elucidate the pathogenic mechanism and status of FPV in Henan, China, to improve control and vaccine development.

## 2. Materials and Methods

### 2.1. Clinical Samples

A total of 82 fecal swab samples were obtained from domestic cats exhibiting clinical symptoms such as fever, vomiting, and diarrhea. These samples were collected from various locations across Henan Province in central China (Appendix A), including Zhengzhou (*n* = 30), Luoyang (*n* = 28), Xinxiang (*n* = 8), Anyang (*n* = 6), Shangqiu (*n* = 5), and Xinyang (*n* = 5). To prepare the samples for analysis, they were subjected to a 10-fold dilution in sterile phosphate-buffered saline (PBS 7.4) supplemented with 100 x antibiotic–antimycotic solution (Gibco, New York, NY, USA). After thorough mixing by vortexing, the diluted samples were centrifuged at 600× *g* for 10 min at 4 °C. The resulting supernatant was filtered through a 0.45 μm pore size membrane filter (Minisart^®^ NML, Sartorius, Germany) and stored in aliquots at −70 °C to preserve viral integrity for subsequent viral DNA extraction and virus isolation procedures.

### 2.2. PCR, Cloning, and DNA Sequencing Assays

In this study, a total of 82 clinical samples were subjected to polymerase chain reaction (PCR) analysis to detect the presence of feline panleukopenia virus (FPV) using a primer targeting an 845-base pair (bp) fragment of the VP2 gene. Viral DNA was extracted from the samples employing the TaKaRa MiniBEST Viral RNA/DNA Extraction Kit (Takara Bio Inc, Shiga, Japan) following the manufacturer’s protocol. The specific concentration (ng/μL) of DNA was detected through the nano-drop ultra-micro spectrophotometer, and the quality of the extracted DNA could be evaluated according to the value of OD_260/280_. The PCR reactions were set up in 50 μL volumes, comprising 10× Taq buffer, 5U of Taq DNA polymerase, and 10 pmol of each FPV-F/R primer (Table 1). Additionally, a synthetic fragment of the FPV-VP2 gene served as a positive control. The thermal cycling conditions consisted of an initial denaturation step at 98 °C for 5 min, followed by 35 cycles of denaturation at 98 °C for 30 s, annealing at 51 °C for 30 s, and extension at 72 °C for 2 min, with a final extension step at 72 °C for 10 min.

Furthermore, for all samples confirmed positive for FPV, the complete NS1 and VP2 genes were amplified using specific primer pairs (NS1-F/R or VP2-F/R) under the same PCR (Table 1) conditions. Subsequently, the PCR products representing the full-length NS1 and VP2 genes were subjected to cloning and sequencing procedures. Gel purification of the PCR products was performed using the QIAquick gel extraction kit, followed by cloning into the pMD19-T vector according to the manufacturer’s instructions. The nucleotide sequences of the cloned genes were determined using an ABI 3730XL DNA Analyzer by Sangon Biotech, Shanghai, China.

### 2.3. Sequence and Phylogenetic Analysis

Phylogenetic analysis was performed based on the NS1 and VP2 nucleotide (2007 and 1755 bp, respectively) and deduced amino acid (669 and 585 aa, respectively) sequences. Sequence alignments and pairwise sequence comparisons were performed using the GENETYX version6 software (Genetyx Corp., Tokyo, Japan); this software facilitated the alignment of genetic sequences and the comparison of sequences between pairs. For phylogenetic analysis, the maximum likelihood method was employed using the MEGA-X software package (version 10.0.5). This analytical tool allowed for the construction of evolutionary trees based on genetic data, with bootstrap values computed from 1000 replicates to assess the robustness of the inferred phylogenetic relationships [17].

### 2.4. Cell Culture and Virus Isolation

Feline kidney (F81) cells were procured from Procell Life Science & Technology Co., Ltd. (Wuhan, China) and maintained in Dulbecco’s modified Eagle’s medium (DMEM; Gibco, Invitrogen, Carlsbad, CA, USA), supplemented with 10% fetal bovine serum (FBS; Gibco, Invitrogen), penicillin (100 U/mL), and streptomycin (100 µg/mL) at 37 °C in a humidified atmosphere containing 5% CO_2_. For virus isolation, the supernatant of the filtered PCR-positive samples was inoculated into F81 cells (80% confluent) at 10% of the total volume of the cell culture medium through a synchronous inoculation method. The cultured cells were maintained in a controlled environment at 37 °C with 5% CO_2_, and they were routinely monitored under a microscope to assess for the presence of cytopathic effects (CPE). Daily observations allowed for the timely detection of any morphological changes indicative of viral infection. Upon reaching 70–80% CPE, signifying substantial viral replication and cell damage, the cultures underwent three cycles of freezing and thawing to release viral particles from the host cells. Subsequently, the lysed cell suspensions were clarified through low-speed centrifugation at 600× *g* for 20 min. This centrifugation step aimed to remove cellular debris and isolate the viral particles from the supernatant. The clarified medium containing the viral particles was then frozen and stored at −80 °C to maintain the integrity of the viral material for further testing and analysis [10]. After two or three passages, FPVs were titrated using a 50% tissue culture infective dose (TCID_50_) in 96-well cell culture plates (SPL Life Sciences). The TCID_50_ was determined in triplicate using the method detailed by Reed and Muench. Cells were seeded in 96-well plates and cultured in F81 cells supplemented with 8% FBS until approximately 90% (3.5 × 10^5^/wells) confluent. The FPV isolate generation cell cultures were each diluted 10-fold in a serial dilution medium (10^−1^–10^−10^). Then, supernatant was removed and then incubated with 100 µL suspensions of each dilution of these isolates. The cells in the control group were inoculated only with medium (100 µL/well). After incubation at 37 °C for 1 h, the F81 cells supplemented with 4% FBS were used to maintain normal growth at 37 °C and 5% CO_2_ for three days.

### 2.5. Animal Experiments

Healthy 8-week-old Dragon Li cats (*n* = 6) were purchased from a pet market in China. These animals were subjected to a week-long period of adaptation. Prior to commencement, anal swabs and serum samples were collected and tested. The serological assays showed that the animals were negative for FPV, *feline coronavirus* (FCoV), feline calicivirus (FCV), and feline herpesvirus (FHV). The six cats were randomly divided into two groups. The first group was orally inoculated with the virus at 2 mL/cat (virus titer was 1.5 × 10^6^ TCID_50_/mL, and 10 mL NaHCO_3_ solution was orally administered before inoculation), and the control group was inoculated with an equal volume DMEM culture solution. The status of the experimental cats was observed every day after inoculation. Postmortem autopsies of the deceased animals were performed to observe the pathological changes in each tissue. A 10% formalin solution was used for fixation, HE staining was performed, and pathological sections were made. Then, the viral load of tested organs (heart, liver, spleen, lung, kidney, small intestine, muscle, stomach, brain) was detected by absolute quantitative real-time PCR, and the target fragment, amplified by FPV quantitative primers (Table 1), was inserted into the pMD19-T cloning vector to construct the recombinant plasmid [18]. All animal tests were approved by the Animal Institution Review Committee and the Use Committee of Henan University of Science and Technology (HAUST-2022-13480102).

## 3. Results

### 3.1. Sample Identification

The 82 samples underwent PCR amplification using FPV primers, resulting in the identification of 25 FPV-positive samples, corresponding to a detection rate of 30.49% (Table 2). Following this, full-length sequencing of the VP2 and NS1 genes was conducted on the FPV-positive samples. Specifically, amplification of the full-length sequences yielded 11 VP2 gene sequences and 21 NS1 gene sequences (Table 3, Appendix A).

### 3.2. Sequence Analysis

The full-length VP2 and NS1 sequences were submitted to the GenBank database under the accession numbers shown in Table 2. The VP2- and NS1-encoding genes of all samples were analyzed and compared with reference isolates (Appendix A). Nucleotide sequence alignment of the VP2-encoding gene of 11 isolates showed high similarity with 44 reference VP2 isolates (including both FPV and CPV isolates). The results showed that the nucleotide homology between the 11 isolates and the reference strains ranged from 97.6% to 100%, while homology between the 11 isolates ranged from 97.6% to 100% (Appendix A). The VP2 amino acid sequences of the 11 isolated strains were compared with the VP2 amino acid sequences of 44 isolates downloaded from the NCBI database. The results showed that the amino acid homology between the 11 isolates and the reference strains ranged from 96.1% to 100%, while the homology between the 11 isolates ranged from 96.1% to 100% (Appendix A). The results showed that the amino acid sequences of Luoyang-13 and Zhengzhou-26 were identical to CPV reference strains at positions 80 (R), 93 (N), 103 (A), 323 (N), 564 (S), and 568 (G), but differed from other isolates and FPV reference strains at positions 80 (K), 93 (K), 103 (V), 323 (D), 564 (N), and 568 (A) (Table 4).

The alignment of NS1 nucleotide sequences of the 21 isolates showed that the nucleotide homology between the 21 isolates and the reference strains ranged from 98.2% to 100%, while the homology between the 21 isolates ranged from 98.6% to 100% (Appendix A). The 21 NS1 amino acid sequences were compared with the NS1 amino acid sequences of 38 isolates (including both FPV and CPV isolates) of FPV published on the NCBI website. The results showed that the amino acid homology between the 21 isolates and the reference strains ranged from 97.6% to 100%, while the homology between the 21 isolates ranged from 98.1% to 100% (Appendix A). The amino acid sequences of Luoyang-13 and Zhengzhou-26 were identical to those of CPV reference strains at positions 23 (N), 247 (Q), 443 (I), 545 (V), and 596 (V), but differed from other isolates and FPV reference strains at positions 23 (D), 247 (H), 443 (V), 545 (E), and 596 (L) (Table 5).

### 3.3. Phylogenetic Analysis

For the VP2-encoding gene, nucleotide sequences of 11 samples and sequence information of 44 selected reference strains were included in the phylogenetic analysis (Appendix A). The results showed that the VP2 nucleotide sequences of the 44 selected parvovirus reference strains were clearly divided into four subgroups: G1 (G1-A, G1-B), G2, G3, and CPV (Figure 1), while nine of the 11 isolates from this study clustered with the G1 group. Eight of the isolates were closely related to local FPV isolates from Shanghai (MW659466) and Beijing (MT270581, MK266797). The Luoyang-13 and Zhengzhou-26 strains fell within the CPV group. It is worth noting that the FPV standard [19] strain CU-4 (M38246) and the US Pfizer vaccine strain (EU498681) belong to the G3 group and are distantly related to the Henan local isolates.

For the NS1-encoding gene, nucleotide sequences of 21 samples were included in the phylogenetic analysis (Figure 2), along with sequence information of 38 selected reference strains (Appendix A). The results showed that the genetic evolution of the NS1 nucleotide sequence of the 38 selected parvovirus reference strains showed obvious G1, G2, G3, and CPV subgroups. The isolates belonged to the G1 group and were closely related to the Chinese isolates (MW659466) and (MN908257), and a Hefei local isolate (MT614366). Moreover, the Luoyang-13 and Zhengzhou-26 isolates both fell within the CPV group. It is worth noting that the Henan local isolate was not in the same group as the FPV standard strain CU-4 (M38246), and that they were only distantly related.

### 3.4. Virus Isolation and Propagation

The endemic FPV strains from Henan Province were mainly from group G1. In this study, the isolate Luoyang-01 (subsequently referred to as FPV-LY), a representative endemic isolate in Henan Province, was selected for the following study. The FPV-LY samples were isolated from the PCR-positive sample, and the isolate was confirmed by diagnostic RT-PCR. Then, F81 cells were inoculated with the FPV-LY isolate and blind transmission to the fifth generation, and CPE was observed in the F81 cells infected with the FPV-LY isolate. The CPE of the isolate manifested obvious pathological phenomena such as cell rounding, wire drawing, and wrinkling (Figure 3). The number of wells with CPE at each dilution was recorded after continuous observation for five days. The virus titer determined by the Reed–Muench method was 1.5 × 10^6^ TCID_50_/mL.

### 3.5. Experimental Infection in Cats

The FPV-LY-inoculated cats began to show symptoms such as vomiting, depression, loss of appetite, and diarrhea seven days after the challenge, and the course of the disease lasted for seven to 12 days. All the FPV-LY-inoculated cats died, while all the cats in the control group survived. Routine blood tests were performed after clinical symptoms appeared on the seventh day after the challenge, and the relative change value of the blood routine index was calculated. Compared with the blood routine index before the challenge, the number of white blood cells, lymphocytes, neutrophils, eosinophils, platelets, and platelet count decreased by 31.00%, 12.82%, 50.52%, 31.43%, 71.04%, and 86.79%, respectively, in the FPV-LY-inoculated group seven days after challenge (Figure 4A). Animals in the challenged group died after 10 to 13 days, with feces around the anus, congestion of the mesentery, hemorrhagic spots in the intestinal wall, and thinning of intestinal segments, the duodenum and rectum being the most obvious. The HE-staining showed no pathological changes in the control group (Figure 4B), but the results of the challenged group showed obvious intestinal villi rupture, shedding, necrosis, and other histopathological changes (Figure 4C). Viral replication was detected in all tissues (brain, heart, liver, spleen, lung, kidney, stomach, small intestine) of the experimental group (Appendix A), after the cats were euthanized. Here, viral replication was the highest in the small intestine (Figure 4D).

## 4. Discussion

FPV is highly contagious and is one of the major pathogens that seriously harm all members of the cat species [20]. Although FPV infection occurs in all countries in the world, the infection rate varies greatly among countries, for example, Australia (29.9%), Italy (45.78%), Portugal (58.0%), and Japan (28.4%) [2,4,6,21]. While there has been limited research conducted on the molecular epidemiology of FPV in China, critical knowledge gaps persist regarding the prevalence of FPV in Henan Province and the molecular characteristics of the circulating strains. The scarcity of studies on this topic underscores the need for comprehensive investigations to elucidate the epidemiological patterns and genetic diversity of FPV in Henan Province, central China. Understanding the prevalence and molecular features of circulating FPV strains is essential for informing public health strategies, veterinary interventions, and the development of effective control measures to mitigate the spread of FPV infections in domestic cat populations. By addressing these knowledge gaps, researchers can contribute to the advancement of our understanding of FPV transmission dynamics and evolution, ultimately aiding in the prevention and management of FPV-associated diseases [22]. Therefore, identifying the molecular genetic characteristics of FPV in domestic cats is of great guiding significance for understanding the prevalence and genetic evolution of FPV, and developing targeted vaccines.

Eleven full-length VP2 gene sequences and twenty-one full-length NS1 gene sequences were obtained from twenty-five FPV-positive samples, indicating that the partial viral DNA in these samples may be incomplete, which may be related to the storage method and time of the samples. The VP2 protein forms most of the capsid structure of the FPV virus and determines the antigenic characteristics and host range of the virus strain [23]. Loop1–Loop4 in the amino acids of VP2 protein constitute five main cyclic tertiary structure regions. As previously reported, four loop regions—loop1 (i.e., V84-D99), loop2 (i.e., R216-G235 and H222-T228), loop3 (i.e., P295-I306 and A300-F303), and loop4 (i.e., Y409-Y444, N421-N428, and T433-N443)—play an important role in the interactions of viruses and host cell surface receptors [24]. Loop1/2/4 not only constitutes the apical part of the VP2 protein folding unit, but also constitutes antigen site A together with the 93, 222, 224, and 426 amino acid sites. Loop3 not only forms the shoulder of the protein folding unit, but also forms antigenic site B together with amino acid sites 299, 300, and 302 [25,26].

The high homology observed in our study lays the foundation for research on FPV/CPV subunit vaccines and nucleic acid vaccines, which could potentially aid in the development of effective preventive measures against FPV. Further analysis revealed these mutations at key sites on the FPV VP2 gene, including positions 80, 85, 87, 93, 101, 103, 232, 297, 299, 300, 305, 323, 426, 564, and 568 amino acids, as identified by previous studies [27,28]. In addition, recent studies have found that FPV VP2 A91S, L562V, and N375D variants have been detected in certain areas of China [29,30]. FPV is the common ancestor of CPV and MEV, and the host range amino acid sites determining FPV are 80, 93, 103, 323, 564, and 568 [31,32]. Mutations at the 91 amino acid sites are located in Loop1, and mutations at the 232 amino acid sites are located in Loop2. Mutations at these sites may change the folding conformation of VP2 protein and the characteristics of antigen site A, thus affecting the recognition of antigen site A by neutralizing antibodies produced by traditional vaccine strains and avoiding immune surveillance [33]. Amino acid residue 93 is a key site in antigen site A, and its mutation may alter the antigenicity and host range of the virion [34]. Studies have shown that amino acid residues at position 93 of VP2 protein have an impact on CPV’s ability to bind to canine transferrin receptor (TfR) and infect host cells, playing an important role in host range [35]. Amino acid residue 300th is a key site in antigen site B, which may be involved in the production of neutralizing antibodies [30], and this mutation may reduce the effectiveness of neutralizing antibodies produced by vaccine strains. Some studies have shown that the change in amino acid 300th may change the host range and coordinate other mutation sites to change the binding of the virus to the TfR receptor, thus changing the infection efficiency [36]. A large number of mutations were found at 93, and 300 amino acid sites were found in our isolated strains. Since many samples were not known to have been vaccinated, it remains unclear whether the variation at these sites is related to vaccine selection pressure or virus evolution.

Interestingly, none of the 11 VP2 isolates from our study exhibited new mutations at these sites. Notably, the 323rd position on the VP2 gene, which determines the antigen specificity of the virus and can influence viral hemagglutination, showed no new mutations in our samples. However, further investigation is needed to elucidate how amino acid variations at these key sites affect viral antigen structure and replication dynamics. This detailed genetic analysis provides valuable insights into the molecular characteristics of FPV strains circulating in Henan Province and underscores the importance of continuous surveillance to monitor genetic changes in the virus population. Based on VP2 gene phylogenetic analysis, three genetic clusters (G1, G2, and G3) are present around the world. In South Korea, FPVs belonging to both the G1 and G2 clusters were found [37]. The phylogenetic analysis of 44 reference strains and 11 full-length VP2 gene sequences obtained here showed that the nucleotide sequences were obviously divided into four subgroups: G1 (G1-A, G1-B), G2, G3, and CPV, among which 9 of the 11 isolates were in the G1 group. Eight of the isolates were closely related to local FPV isolates from Shanghai (MW659466) and Beijing (MT270581 and MK266797), and the Luoyang-13 and Zhengzhou-26 strains fell within the CPV group. The FPV standard strain CU-4 (M38246) and the US Pfizer vaccine strain EU498681 clustered with the G3 group and are distantly related to the isolated strains, suggesting that FPV mutates and evolves rapidly. The differences in genome sequence identity and genetic branching between the isolates indicate that the vaccine immunity of FPV is poor.

Although current research on the VP2 gene of FPV is relatively sufficient, research on the NS1 gene is not robust enough to understand the corresponding protein structures and functions, and only five amino acid mutation sites have been found. In previous studies, researchers evaluated the function of the NS1 protein and described the potential location of its functional domains: the origin of replication (ORI) binding site (16–275 aa), helicase (299–486 aa), and the transactivation (600–667 aa) functional domains [38]. In this study, the nucleotide and amino acid homology within and between the 21 NS1 full-length gene sequences amplified here and the 38 reference strains were 98.2% to 100% and 97.6% to 100%, respectively, suggesting that the probability of viral genome recombination was low. At the same time, results showed that the NS1 amino acid sequence of CPV and FPV are different at the following sites (CPV→FPV), 23 (N→D), 247 (Q→H), 443 (I→V), 545 (V→E), and 596 (V→L). In addition, the nucleotide homology of the FPV isolates with CPV and MEV strains was more than 99%, and the amino acid homology was more than 97% and 98%, respectively. However, studies have shown that the mutation rate of the NS1 amino acid sequence is higher than that in VP2, indicating that VP2 is relatively stable compared with NS1 in protein function [39]. However, the phylogenetic analysis of the 21 isolates and 38 reference isolates showed that 19 isolates belonged to the G1 group, which is closely related to Chinese isolates (MW659466 and MN908257) and the Hefei local isolate (MT614366), while two of these isolates (Luoyang-13 and Zhengzhou-26 strains) belong to the CPV branch. Thus, the NS1 sequence of parvoviruses is relatively conserved. The FPV gene clusters in some areas of Henan Province have obvious Asian isolates, and are distantly related to the European, the United States, Oceania, and Australian isolates.

Through sequencing analysis and amino acid comparison, the FPV-LY isolate fell within the G1 group and was selected as the local representative strain. An expanded culture of this FPV-LY virus was performed by animal regression tests with a titer of 1.5 × 10^6^ TCID_50_/mL, and the results showed significant pathogenicity. Similarly, FPV isolated from intestinal samples caused high morbidity and mortality in minks [40]. Also, the decrease in white blood cells detected here is consistent with previous studies [41,42]. Moreover, FPV-LY eventually led to the death of all experimental cats with obvious clinical symptoms, where lesions and damage were concentrated in the small intestine, mesentery, spleen, and other organs, similar to the results of a previous study [43]. Finally, the virus content in the tissues and organs was consistent with the results of the distribution of the lesions.

## 5. Conclusions

In summary, 25 FPV-positive samples were identified at a positive rate of 30.49%. Then, 11 VP2 and 21 NS1 full-length gene sequences were generated, and the following homology analysis showed that the amino acid homology of the VP2/NS1 genes with that of the endemic strains and those from abroad were 96.1–100% and 97.6–100%, respectively. The FPV Henan isolates fell within the G1 group, which is distantly related to FPV standard strains and vaccine strains. The FPV-LY isolate was selected as the local representative strain in Henan province, and the results of the animal regression test showed that the strain had strong virulence. These results suggest that novel FPV vaccines may need to be developed due to the constant variation of amino acids during the evolution of viral epidemics.

## Figures and Tables

**Figure 1 vetsci-11-00292-f001:**
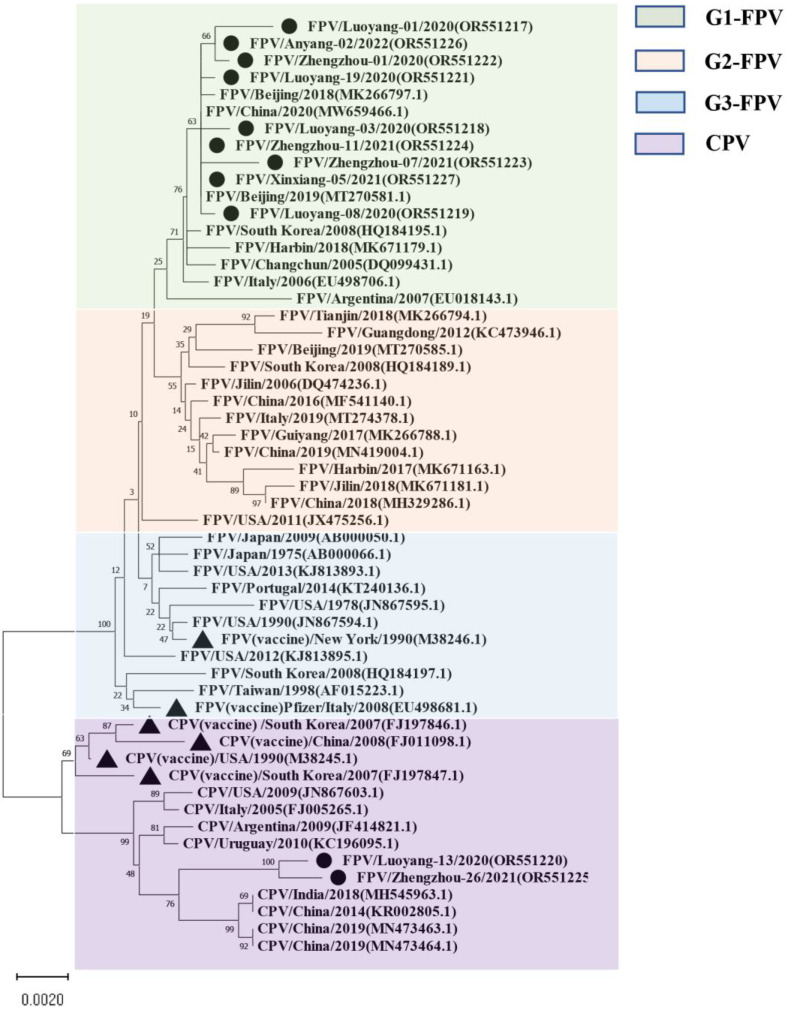
The maximum likelihood phylogenetic tree based on the complete VP2 sequences of FPV and CPV reference strains. Nucleotide sequences were analyzed using MEGA X with a bootstrap test of 1000 replicates (Bootstrap values are shown on branches). The FPV samples can be divided into three groups: the green background is group 1; the orange background is group 2; the blue background is group 3, and the purple background is the reference sequence of CPV. The symbols used to distinguish the different strains are as follows: The black circle (●) indicates isolates from this study, black triangle (▲) indicates the vaccine isolates. Each sequence on the tree is identified by the isolate name, host, country of origin, year of isolation, and GenBank accession number.

**Figure 2 vetsci-11-00292-f002:**
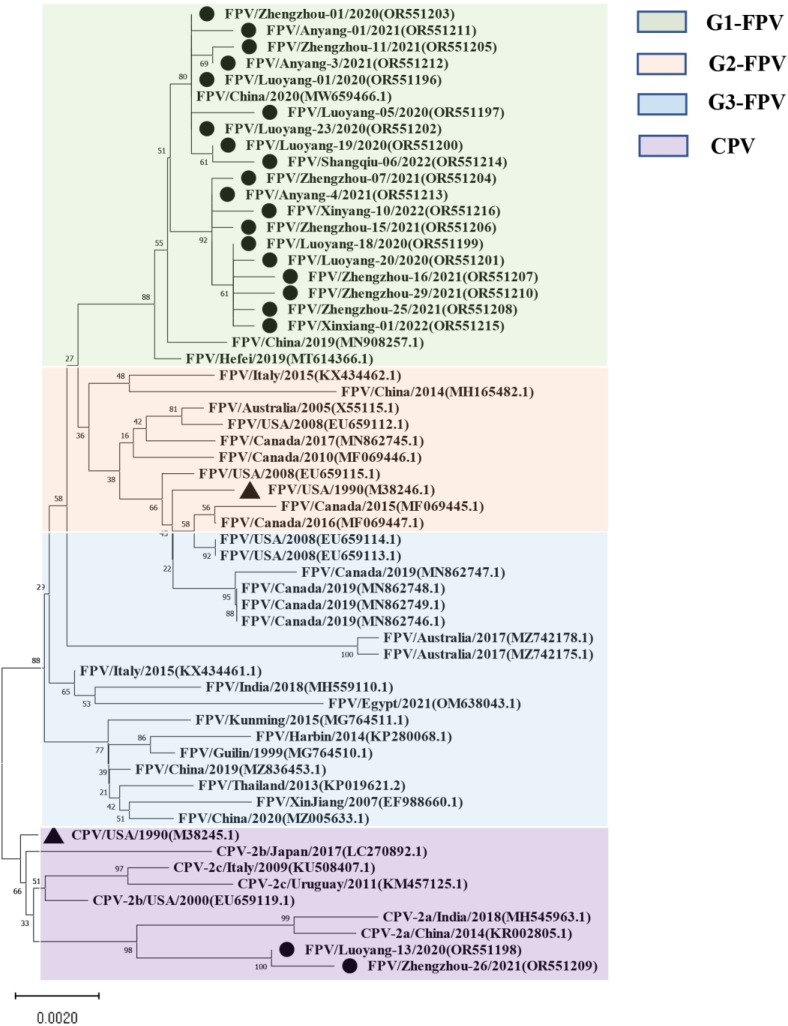
Maximum likelihood tree based on the complete NS1 sequences of FPV and CPV reference sequences. Nucleotide sequences were analyzed with Mega X with a bootstrap test of 1000 replicates (shown on branches). The FPV samples can be divided into three gene groups: the green background is group 1; the orange background is group 2; the blue background is group 3, and the purple background is the reference sequence of CPV. The symbols used to distinguish the different strains are as follows: The red circle (●) indicates isolates from this study, black triangle (▲) indicates the vaccine isolates. Each sequence on the tree is identified by the isolate name, host, country of origin, year of isolation, and GenBank accession number.

**Figure 3 vetsci-11-00292-f003:**
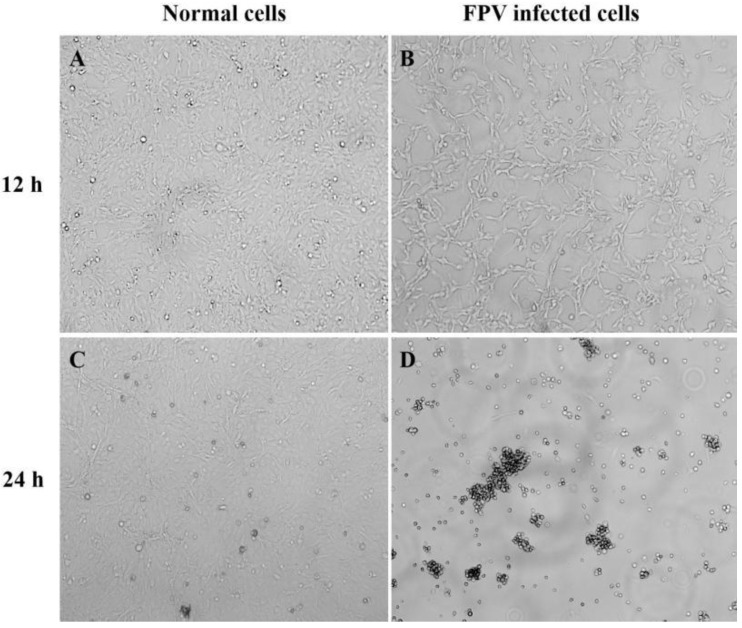
Cytopathic effect (CPE) in FPV-infected F81 cells. Amplification factor 100: (**A**) Non-infected F81 cell culture after 12 h. (**B**) Representative isolate (FPV-LY) infected F81 cells at 12 h post-infection. (**C**) Non-infected F81 cell culture after 24 h. (**D**) Representative isolate (FPV-LY) infected F81 cells at 24 h post-infection.

**Figure 4 vetsci-11-00292-f004:**
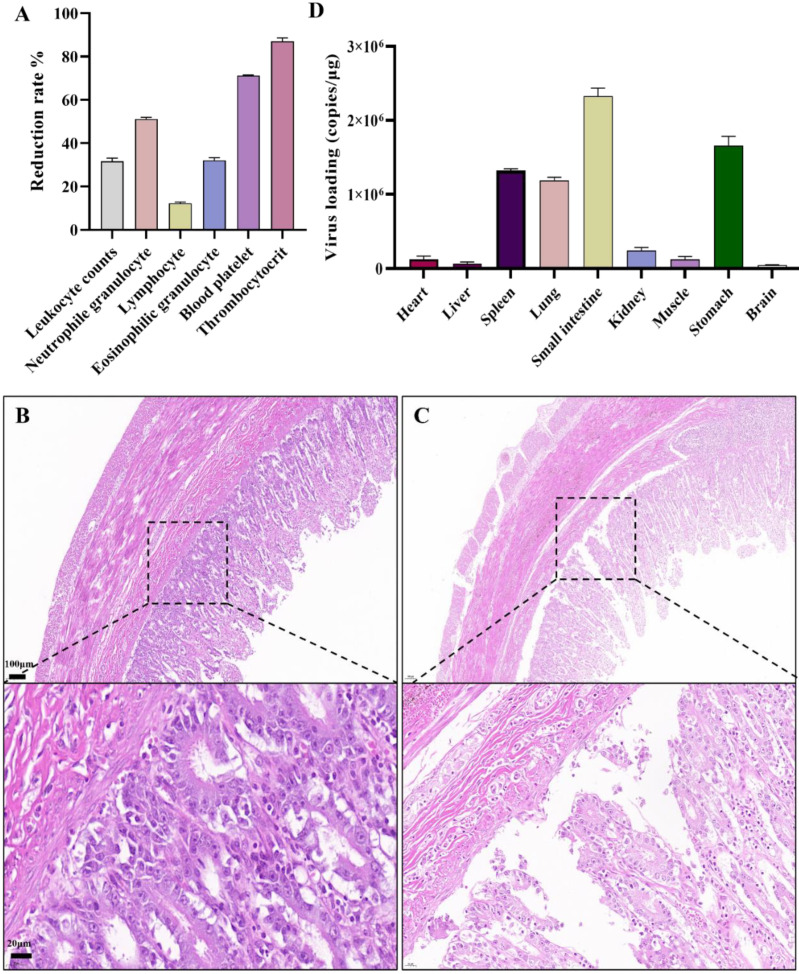
Pathogenicity analysis of FPV in vivo: (**A**) Routine blood test. (**B**) HE-staining in the control group showing no histopathological changes in cat small intestine (100 μm). (**C**) Histopathological changes in cats infected with the FPV-LY isolate. (**D**) Viral loading in tested organ.

**Table 1 vetsci-11-00292-t001:** Primers used for the amplification of the VP2 and NS1 genes of FPV isolated from cats in China.

Name	Sequence (5′-3′)	Location (bp)	Size (bp)	Restriction Enzyme Cutting Site
FPV-F ^a^	GGATGGGTGGAAATCACAGC	2949–2968	845	-
FPV-R	ATAACCAACCTCAGCTGGTC	3775–3794		-
VP2-F	TTGGATCCATGAGTGATGGAGCAGTTC	2781–2799	1755	*Bam*H I
VP2-R	GGAAGCTTTTAATATAATTTTCTAGGTGC	4515–4535		*Hin*d III
NS1-F	CTTTGACTAACCATGTCTGGC	267–293	2007	-
NS1-R	CTTACCTCTCCTGGCTCTCTT	2247–2273		-
qFPV-F	CATTGGGCTTACCACCATTT	851–870	208	-
qFPV-R	AAATGGCCCTTGTGTAGACG	1040–1059		-

^a^ FPV-F/R used for viral screening; VP2-F/R and NS1-F/R were used for amplification for Sanger sequencing; underlined part of sequence means sites recognized by the restriction enzymes in VP2 primers.

**Table 2 vetsci-11-00292-t002:** Detailed description of feline panleukopenia virus (FPV) samples characterized in this study.

Strains	Location	Time	Species	Age (Weeks)	Gender	GenBank Accession Number	
						NS1	VP2
Luoyang-01	Luoyang	Jul-2020	Cat	28	Female	OR551196	OR551217
Luoyang-03	Luoyang	Jul-2020	Cat	35	Male	×	OR551218
Luoyang-05	Luoyang	Jul-2020	Cat	103	Female	OR551197	×
Luoyang-08	Luoyang	Aug-2020	Cat	92	Female	×	OR551219
Luoyang-13	Luoyang	Aug-2020	Cat	47	Female	OR551198	OR551220
Luoyang-18	Luoyang	Aug-2020	Cat	42	Male	OR551199	×
Luoyang-19	Luoyang	Sep-2020	Cat	38	Male	OR551200	OR551221
Luoyang-20	Luoyang	Sep-2020	Cat	146	Female	OR551201	×
Luoyang-23	Luoyang	Oct-2020	Cat	25	Female	OR551202	×
Zhengzhou-01	Zhengzhou	Oct-2020	Cat	56	Male	OR551203	OR551222
Zhengzhou-07	Zhengzhou	Apr-2021	Cat	254	Male	OR551204	OR551223
Zhengzhou-11	Zhengzhou	Apr-2021	Cat	16	Male	OR551205	OR551224
Zhengzhou-15	Zhengzhou	May-2021	Cat	85	Female	OR551206	×
Zhengzhou-16	Zhengzhou	May-2021	Cat	65	Female	OR551207	×
Zhengzhou-25	Zhengzhou	May-2021	Cat	45	Female	OR551208	×
Zhengzhou-26	Zhengzhou	Jun-2021	Cat	35	Female	OR551209	OR551225
Zhengzhou-29	Zhengzhou	Jun-2021	Cat	95	Male	OR551210	×
Anyang-01	Anyang	Jul-2021	Cat	74	Male	OR551211	×
Anyang-02	Anyang	Jul-2021	Cat	15	Male	×	OR551227
Anyang-03	Anyang	Jul-2021	Cat	96	Female	OR551212	×
Anyang-04	Anyang	Jul-2021	Cat	25	Male	OR551213	×
Shangqiu-06	Shangqiu	Mar-2022	Cat	135	Female	OR551214	×
Xinxiang-01	Xinxiang	Jun-2022	Cat	245	Female	OR551215	×
Xinxiang-05	Xinxiang	Jun-2022	Cat	269	Male	×	OR551226
Xinyang-10	Xinyang	Aug-2022	Cat	25	Male	OR551216	×

“×” indicates sequencing failure; NS1, non-structural protein 1; VP2, structural protein.

**Table 3 vetsci-11-00292-t003:** FPV sample source and PCR amplification of the VP2 and NS1 genes.

Origin	Sampling No.	No. of Positive	No. of VP2 Gene Detection	No. of NS1 Gene Detection
Zhengzhou	30	8	4	8
Luoyang	28	8	5	7
Xinxiang	8	4	1	3
Anyang	6	2	1	2
Shangqiu	5	2	0	1
Xinyang	5	1	0	1
Sum	82	25	11	21

**Table 4 vetsci-11-00292-t004:** Deduced amino acid substitutions in the VP2-encoding gene of FPV isolates from dogs and cats.

Virus Strain	GenBank Accession No.	Amino Acid Position								
		80	93	103	297	300	305	323	564	568
FPV(vaccine)/New York/1990	M38246.1	K	K	V	S	A	D	D	N	A
FPV(vaccine)Pfizer/Italy/2008	EU498681.1	K	K	V	S	A	D	D	N	A
CPV(vaccine)/South Korea/2007	FJ197846.1	R	N	A	A	G	Y	N	S	G
CPV(vaccine)/China/2008	FJ011098.1	R	N	A	A	G	Y	N	S	G
CPV(vaccine)/USA/1990	M38245.1	R	N	A	A	G	Y	N	S	G
CPV(vaccine)/South Korea/2007	FJ197847.1	R	N	A	A	G	Y	N	S	G
FPV/China/2020	MW659466.1	K	K	V	S	A	D	D	N	A
FPV/Beijing/2018	MK266797.1	K	K	V	S	A	D	D	N	A
FPV/Beijing/2019	MT270585.1	K	K	V	S	A	D	D	N	A
FPV/South Korea/2008	HQ184189.1	K	K	V	S	A	D	D	N	A
FPV/Luoyang-01/2020	OR551217	K	K	V	S	A	D	D	N	A
FPV/Luoyang-03/2020	OR551218	K	K	V	S	A	D	D	N	A
FPV/Luoyang-08/2020	OR551219	K	K	V	S	A	D	D	N	G
FPV/Luoyang-13/2020	OR551220	R	N	A	A	G	Y	N	S	G
FPV/Luoyang-19/2020	OR551221	K	K	V	S	A	D	D	N	A
FPV/Zhengzhou-01/2020	OR551222	K	K	V	S	A	D	D	N	A
FPV/Zhengzhou-07/2021	OR551223	K	K	V	S	A	D	D	N	A
FPV/Zhengzhou-11/2021	OR551224	K	K	V	S	A	D	D	N	A
FPV/Zhengzhou-26/2021	OR551225	R	N	A	A	G	Y	N	S	G
FPV/Anyang-02/2022	OR551226	K	K	V	S	A	D	D	N	A
FPV/Xinxiang-05/2021	OR551227	K	K	V	S	A	D	D	N	A

Note: The grey background in the table represents the isolates in this study.

**Table 5 vetsci-11-00292-t005:** Deduced amino acid substitutions in the NS1-encoding gene of FPV isolates from dogs and cats.

Virus Strain	GenBank Accession No.	Amino Acid Position									
		23	60	247	356	443	545	572	596	624	667
FPV/New York/1990	M38246.1	D	I	H	N	V	Q	E	V	N	L
FPV/China/2019	MN908257.1	D	I	H	N	V	E	E	L	N	L
FPV/Hefei/2019	MT614366.1	D	I	H	N	I	E	E	L	N	L
FPV/USA/2008	EU659112.1	N	I	H	N	I	Q	E	V	N	L
FPV/Italy/2015	KX434461.1	D	I	H	N	V	E	E	V	N	L
FPV/Canada/2017	MN862745.1	D	I	H	N	V	Q	E	V	N	L
FPV/China/2020	MW659466.1	D	I	H	N	V	E	E	L	N	L
CPV/USA/1990	M38245.1	N	I	Q	N	I	E	E	V	N	L
CPV-2a/India/2018	MH545963.1	N	I	Q	N	I	E	E	V	N	L
CPV-2b/Japan/2017	LC270892.1	N	I	Q	N	I	E	K	V	N	L
FPV/Luoyang-01/2020	OR551196	D	I	H	N	V	E	E	L	N	L
FPV/Luoyang-05/2020	OR551197	D	I	H	N	V	E	E	L	N	L
FPV/Luoyang-13/2020	OR551198	N	V	Q	N	I	V	E	V	N	L
FPV/Luoyang-18/2020	OR551199	D	I	H	N	V	E	E	L	N	L
FPV/Luoyang-19/2020	OR551200	D	I	H	N	V	E	E	L	N	L
FPV/Luoyang-20/2020	OR551201	D	I	H	N	V	E	E	L	N	L
FPV/Luoyang-23/2020	OR551202	D	I	H	N	V	E	E	L	N	L
FPV/Zhengzhou-01/2020	OR551203	D	I	H	N	V	E	E	L	N	L
FPV/Zhengzhou-07/2021	OR551204	D	I	H	N	V	E	E	L	N	L
FPV/Zhengzhou-11/2021	OR551205	D	I	H	N	V	E	E	L	N	L
FPV/Zhengzhou-15/2021	OR551206	D	I	H	N	V	E	E	L	N	L
FPV/Zhengzhou-16/2021	OR551207	D	I	H	N	V	E	E	L	N	L
FPV/Zhengzhou-25/2021	OR551208	D	I	H	N	V	E	E	L	N	L
FPV/Zhengzhou-26/2021	OR551209	N	V	Q	N	I	V	E	V	N	L
FPV/Zhengzhou-29/2021	OR551210	D	I	H	N	V	E	E	V	N	L
FPV/Anyang-01/2021	OR551211	D	I	H	N	V	E	E	L	N	L
FPV/Anyang-3/2021	OR551212	D	I	H	N	V	E	E	L	N	L
FPV/Anyang-4/2021	OR551213	D	I	H	N	V	E	E	L	N	L
FPV/Shangqiu-06/2022	OR551214	D	I	H	N	V	E	E	L	N	L
FPV/Xinxiang-01/2022	OR551215	D	I	H	N	V	E	E	L	N	L
FPV/Xinyang-10/2022	OR551216	D	I	H	N	V	E	E	L	N	L

Note: The grey background in the table represents the isolates in this study.

## Data Availability

The datasets presented in this study can be found in in the article/Appendix A.

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
