# Peer review of "Molecular Characterization of Feline Parvovirus from Domestic Cats in Henan Province, China from 2020 to 2022"

_vetsci, 2024, doi:10.3390/vetsci11070292_

Round 1
Reviewer 1 Report
Comments and Suggestions for Authors
I would like to thank the Editor for the opportunity to review this manuscript. Paper entitled – ‘ Molecular characterization of feline parvovirus from domestic cats in Henan Province, China from 2020 to 2022’ presents the study concerning quite common, dangerous and interesting pathogen. Evolutionary history of Parvoviridae is great example how single mutation matters and how it may expand the host range. Therefore, tracing and analyzing of molecular polymorphism and investigating the effects of changes in nucleotide and aa sequence on the virulence of parvoviruses are very important.
Paper prepared by Yu et al., seems to embrace this topic and it worthy to consider to be published. At least, at first glance. Nevertheless, when we look more carefully, there are some drawbacks, and questions. I have read the paper and have some thoughts to share. Here below, you can find the detailed list of comments. Hope that you find them as useful in enhancing the quality of paper.
Title – I have no comments to title it reflects the content of the paper.
Abstract – Generally well-written but I have two small suggestions.
Line 17 – Term - full-length primers - What does it mean full-length primers ? I guess you mean primers pairs amplifying whole VP2 and NS1 genes ? If so, specify it.
Line 26 – I think that you should mention about the titer of virus causing this effect.
Introduction – In my opinion Authors should also mention about few things in the introduction.
Line 42 - I would like to see one more statement concerning CPV-2. Please add information about evolutionary relation between FPV and CPV.
Lines 42-43 - ‘However, CPV-2 can only replicate in vivo in canids’ – it is not quite precise statement, to be honest some types of CPV-2 may replicate in feline hosts. Therefore, the information about host range should be explained in more detailed way.
Line 45 - One thing, Parrish (1990) put this name in quotation marks and it does matter as this expression is not quite literal. I suggest also put this term in quotation marks.
Lines 47-49 – Good point.
Line 62 - There is also at least one more point. Notice, that constant evolution of FPV may hinder the molecular diagnostics, as mutation/polymorphism may occur in primer binding sites. I think that it is worthy to mention about it. Better picture of molecular epidemiology vastly facilitates to be up-to-date in molecular testing.
Materials and methods – This section needs some additional information to make the whole procedure repeatable and to shed more light on some questions.
Line 76 – Provide more details about animals - such as age, gender, etc. Moreover, do you have any data about vaccination status of the tested animals ? In conclusion you claim that there is necessity of new vaccines and the present one may be ineffective, how many of the tested cats were vaccinated and what kind of vaccines have been administrated ?
Line 76 - I would suggest to provide map presenting mentioned area even as the supplementary figure.
Line 86 - Why did you decide to amplify relatively long sequences in diagnostic PCR ?
In case of challenging material such as fecal swab, you may obtain degraded DNA and obtaining longer amplicons may be difficult. As a result you may get false negative results.
Line 87 - Did you verify the quantity and quality of DNA after extraction ?
Line 88 - Understand, that you followed the manufacturer protocol, but i would like to see some additional details concerning extraction, such as elution volume.
Line 89 - Fecal swabs are quite challenging material, as such material may contain PCR inhibitors. Did you use any internal control to verify inhibition of PCR?
Table 1 - I suppose that you pointed the sites recognized by the restriction enzymes in VP2 primers (underlined part of sequence), if so, please add this information in table description.
Line 111 – There is some contradiction, here you wrote that you used ML method in phylogenetic analysis, but in figures description there is NJ, where is the true version ?
Line 136 - Were these animals subjected any quarantine or acclimatization period ?
Line 147 - What means 'each organ' - specify.
Line 148 - What was the further application of this plasmid ? Was it subjected to sequencing ?
Results – Some issues need clarification
Line 154 – font of ‘25/82’
Table S1 – Consider if it is not worthy to include the Table S1 in the main manuscript.
Table 2 – Remove |from the table description
Table S2 and S3 - Maybe 'isolation year' instead of 'separation of the year' ?
Line 167 – Exactly, the same values 11 isolates vs reference isolates ?
Fig S1 - You should include some description on the figures as they are not self-explanatory. For example mention if the picture concerns the NS1 or VP2 gene and if presented homology concerns nucleotide or aa sequence ? Is it possible to highlight your strains in the phylogenetic tree included in the figure ? it will be far easier to trace and compare results.
Line 176 (and the whole paragraph) - 11 or 21 ? According to the results there were 21 positively sequenced samples for NS1 sequence. Also 21 sequences are presented in supplementary figures.
Line 180 - Explain this issue. When I’m checking Table S3, I see over 44 sequences (not 44 as you claim) and 31 belong to FPV, whereas rest is noted as CPV.
Table 3 - I would suggest to make the table more clear and easy to trace. Have you consider to apply some shadowing to distinguish vaccine strains and your unique strains such as Luoyang-13 ?
Line 190 - Some branches display very low bootstrap values. Especially in G1 group. Could you explain it ? Have you consider to edit your tree to display the separation of branches which display higher values of bootstrap ?
Figure 1 – I’m confused as you mentioned in material and methods section that you applied the ML method, whereas in figure description I see NJ, which one was used to design the phylogentic tree ?
Are you sure, that isolates which are kind of outgroup such as FPV/USA/2011 (JX475256.1) may be counted without hesitation to the particular groups ? In this case to G3 ? Similar
situation in case of FPV/Italy/2006 and FPV/Argentina/2007 why did you include them to G2? Another issue, I see that you phylogenetic tree contains not much the newest sequences. How did you choose the sequences to your alignment, there were any criteria ?
Line 260 - Could you provide the average viral load for each tissue ? There is not so easy to get values from the figure (I know that figure just present the scale of differences) but I would be glad to see the values.
Line 266 – better in ‘tested organs’ than ‘each organ’
Discussion – in my opinion this is the poorest part of the paper. Mostly, this section is the repetition of the results section and it should be edited thoroughly. Authors should rewrite this whole section to discuss their results and compare them with the other studies.
Line 282 – ‘Stable mutations’ is not the best expression as mutations are far from be stable.
Line 287 - Provide some reference
Lines 275-300 - This is mostly repeated results section. You should discuss your results more extensively , comparing your results with observations of other research teams.
Lines 314-316 - it is far more complex issue. Note, that not each change in nucleotide sequence is followed by the sequence in aa sequence. What about synonymous mutation ? They may not change aa sequence, but may be involved for example in binding TFs. Discuss this issue more extensively.
Line 320 - Several lines above you you put the statement that mutation rate in NS1 is higher, here you claimed that NS1 is conserverd. Explain it and support the statement with proper reference.
Line 321 - explain what does mean "Asian characteristics' ?
Line 323-325 - General statement that not quite stem from your paper.
Lines 329-331 - This is a sample of what discussion should be. Mostly this section is repetition of results section. As I mention above, compare your results with other studies.
Conlusion – Final statement (Lines 345-346) are not quite justified, as we know nothing about the vaccination status of the tested animals.
Decision – The paper presents current and interesting topic. However there are some doubts which should be explained. Paper suffer from the some flaws which make it impossible to accept it in present form. I recommend major revision to give the Author chance to address my comments and to present their point of view. Hope, you find my suggestions helpful.

Author Response
Response to reviewers
We would like to thank you for your helpful comments and suggestions, which were valuable to improve the quality of our manuscript. We have extensively and carefully revised the entire manuscript based on your review. A revised manuscript with the corrections highlighted using blue mark has been prepared. The reviewer’s comments are shown below followed by our pointwise responses.
Reviewer 1
- Abstract – Generally well-written but I have two small suggestions.
Line 17 – Term - full-length primers - What does it mean full-length primers? I guess you mean primers pairs amplifying whole VP2 and NS1 genes? If so, specify it.
Answer: As suggested, we have revised the description of primers in the manuscript.
Line 26 – I think that you should mention about the titer of virus causing this effect.
Answer: As suggested, we have added information about the titer of virus causing CPE.
- Introduction– In my opinion Authors should also mention about few things in the introduction.
Line 42 - I would like to see one more statement concerning CPV-2. Please add information about evolutionary relation between FPV and CPV.
Answer: As suggested, we have added information about the evolutionary relation between FPV and CPV.
Lines 42-43 - ‘However, CPV-2 can only replicate in vivo in canids’ – it is not quite precise statement, to be honest some types of CPV-2 may replicate in feline hosts. Therefore, the information about host range should be explained in more detailed way.
Answer: Thank you for your reminder. As suggested, we have added information about host range in detail.
Line 45 - One thing, Parrish (1990) put this name in quotation marks and it does matter as this expression is not quite literal. I suggest also put this term in quotation marks.
Answer: As suggested, we have modified this sentence in the manuscript.
Lines 47-49 – Good point.
Answer: Thank you for your recognition.
Line 62 - There is also at least one more point. Notice, that constant evolution of FPV may hinder the molecular diagnostics, as mutation/polymorphism may occur in primer binding sites. I think that it is worthy to mention about it. Better picture of molecular epidemiology vastly facilitates to be up-to-date in molecular testing.
Answer: As suggested, we have added information about that constant evolution of FPV may hinder the molecular diagnostics in the manuscript.
- Materials and methods– This section needs some additional information to make the whole procedure repeatable and to shed more light on some questions.
Line 76 – Provide more details about animals - such as age, gender, etc. Moreover, do you have any data about vaccination status of the tested animals ? In conclusion you claim that there is necessity of new vaccines and the present one may be ineffective, how many of the tested cats were vaccinated and what kind of vaccines have been administrated?
Answer: Thanks for your question. As suggested, we have provided more detail information about sampling animals in Table 2. We are sorry that we did not record the vaccination status when we collected the samples, so we are unable to provide this information. In the following epidemiological studies, we will definitely pay more attention and thank you again for your proposal.
Line 76 - I would suggest to provide map presenting mentioned area even as the supplementary figure.
Answer: As suggested, we have prepared a map presenting mentioned area even as the supplementary figure (Fig. S1).
Line 86 - Why did you decide to amplify relatively long sequences in diagnostic PCR ?
In case of challenging material such as fecal swab, you may obtain degraded DNA and obtaining longer amplicons may be difficult. As a result you may get false negative results.
Answer: Thanks for your good questions. The specificity and accuracy can be guaranteed by using the sequence of relatively long sequences. Short sequences are prone to false positives and require secondary testing. We decide to amplify relatively long sequences is based on the result of our comprehensive specificity, accuracy and many other considerations.
Line 87 - Did you verify the quantity and quality of DNA after extraction ?
Answer: Yes, we verified the quantity and quality of DNA after extraction using nano-drop ultra-micro spectrophotometer. For samples with poor quality, we will extract them again.
Line 88 - Understand, that you followed the manufacturer protocol, but i would like to see some additional details concerning extraction, such as elution volume.
Answer: Thanks for your question. We use 45 µL in DNA elution buffer according to the information provided by the kit.
Line 89 - Fecal swabs are quite challenging material, as such material may contain PCR inhibitors. Did you use any internal control to verify inhibition of PCR?
Answer: You're right, the components in the fecal swabs are complex and likely contain PCR inhibitors, which we neglected during the previous experiment, and thanks for your good suggestion, we will use internal control to verify inhibition of PCR in future studies.
Table 1 - I suppose that you pointed the sites recognized by the restriction enzymes in VP2 primers (underlined part of sequence), if so, please add this information in table description.
Answer: As suggested, we have added this information in table description.
Line 111 – There is some contradiction, here you wrote that you used ML method in phylogenetic analysis, but in figures description there is NJ, where is the true version ?
Answer: Thanks for your question. We used ML method in phylogenetic analysis. We have revised the figures description in the manuscript.
Line 136 - Were these animals subjected any quarantine or acclimatization period ?
Answer: Yes, these animals were subjected to a week-long period of adaptation.
Line 147 - What means 'each organ' - specify.
Answer: As suggested, we have added more detail information about ‘each organ’ in the manuscript.
Line 148 - What was the further application of this plasmid ? Was it subjected to sequencing ?
Answer: Thanks for your question. The constructed plasmid will be used as the standard curve of real-time quantitative PCR to detect the tissue samples we collected in the following study. The plasmid has been sequenced to ensure the accuracy of the sequence of cloned target gene.
- Results– Some issues need clarification
Line 154 – font of ‘25/82’
Answer: As suggested, we have made modification in the manuscript.
Table S1 – Consider if it is not worthy to include the Table S1 in the main manuscript.
Answer: Thanks for your suggestion, it’s better to include Table S1 in the main manuscript. We have made revision in the manuscript.
Table 2 – Remove |from the table description
Answer: As suggested, we have made revision in the manuscript.
Table S2 and S3 - Maybe 'isolation year' instead of 'separation of the year' ?
Answer: As suggested, we have made revision in the manuscript.
Line 167 – Exactly, the same values 11 isolates vs reference isolates ?
Answer: Yes, sequence analysis showed the same results.
Fig S1 - You should include some description on the figures as they are not self-explanatory. For example mention if the picture concerns the NS1 or VP2 gene and if presented homology concerns nucleotide or aa sequence? Is it possible to highlight your strains in the phylogenetic tree included in the figure? it will be far easier to trace and compare results.
Answer: As suggested, we have revised all supplementary figures.
Line 176 (and the whole paragraph) - 11 or 21? According to the results there were 21 positively sequenced samples for NS1 sequence. Also 21 sequences are presented in supplementary figures.
Answer: Sorry for the confusion caused by our unclear description. For VP2 gene, 11 isolates were tested positive, for NS1 gene, 21 isolates were tested positive. We have revised this problem in the manuscript.
Line 180 - Explain this issue. When I’m checking Table S3, I see over 44 sequences (not 44 as you claim) and 31 belong to FPV, whereas rest is noted as CPV.
Answer: Sorry for the error in the manuscript. We wrote 44 reference strains in the manuscript, but 46 reference strains are listed in the table, and we have removed duplicate two strains. Furthermore, due to the high homology of FPV and CPV, our reference strains contain some CPV strains, which have been described in the manuscript.
Table 3 - I would suggest to make the table more clear and easy to trace. Have you consider to apply some shadowing to distinguish vaccine strains and your unique strains such as Luoyang-13 ?
Answer: As suggested, we used shadows in the table to distinguish vaccine strains and our isolated strains.
Line 190 - Some branches display very low bootstrap values. Especially in G1 group. Could you explain it? Have you consider to edit your tree to display the separation of branches which display higher values of bootstrap?
Answer: Indeed, some branches display very low bootstrap values, which may be due to improper selection of some coefficients during our analysis. We re-analyzed the sequences and have got higher values of bootstrap.
Figure 1 – I’m confused as you mentioned in material and methods section that you applied the ML method, whereas in figure description I see NJ, which one was used to design the phylogentic tree?
Are you sure, that isolates which are kind of outgroup such as FPV/USA/2011 (JX475256.1) may be counted without hesitation to the particular groups? In this case to G3? Similar situation in case of FPV/Italy/2006 and FPV/Argentina/2007 why did you include them to G2? Another issue, I see that you phylogenetic tree contains not much the newest sequences. How did you choose the sequences to your alignment, there were any criteria?
Answer: First of all, thank you very much for your valuable questions. Perhaps our expression is not accurate enough. We used ML method in phylogenetic analysis. We have revised the figures description in the manuscript. The actual situation is that we used GENETYX software to compare the nucleotide sequence of VP2 with the amino acid sequence, and MEGA-X software was used to make the evolutionary tree of the sequence. Because we think the MEGA-X is more aesthetically pleasing, that's just our opinion. Secondly, there are some problems in the grouping of FPV/Italy/2006 and FPV/Argentina/2007 sequences. In view of the questions you raised, we have thought through the numerical values and found that these two sequences are not suitable for such grouping, so as to better understand your meaning. Therefore, thank you again for your questions and patience.
Line 260 - Could you provide the average viral load for each tissue? There is not so easy to get values from the figure (I know that figure just present the scale of differences) but I would be glad to see the values.
Answer: Thanks for your questions. The virus loading of tested tissues were shown as follows:
|
Tested organs |
Virus loading (copies/μg) |
||
|
No. 1 |
No. 2 |
No. 3 |
|
|
Liver |
58026 |
40212 |
91986 |
|
Spleen |
1300124 |
1336605 |
1336997 |
|
Lung |
1216069 |
1212728 |
1143822 |
|
Small intestine |
2202983 |
2380581 |
2401285 |
|
Kidney |
272930 |
261592 |
197859 |
|
Muscle |
167774 |
92758 |
99970 |
|
Stomach |
1802983 |
1580581 |
1601285 |
|
Brain |
51321 |
44155 |
41460 |
Line 266 – better in ‘tested organs’ than ‘each organ’
Answer: As suggested, we have made revision in the manuscript.
- Discussion – in my opinion this is the poorest part of the paper. Mostly, this section is the repetition of the results section and it should be edited thoroughly. Authors should rewrite this whole section to discuss their results and compare them with the other studies.
Answer: As suggested, we have revised the entire discussion section.Thank you for your suggestion.
Line 282 – ‘Stable mutations’ is not the best expression as mutations are far from be stable.
Answer: As suggested, we have made revision in the manuscript.
Line 287 - Provide some reference
Answer: As suggested, we have added related reference.
Lines 275-300 - This is mostly repeated results section. You should discuss your results more extensively , comparing your results with observations of other research teams.
Answer: As suggested, we have made revision in the manuscript.
Lines 314-316 - it is far more complex issue. Note, that not each change in nucleotide sequence is followed by the sequence in aa sequence. What about synonymous mutation? They may not change aa sequence, but may be involved for example in binding TFs. Discuss this issue more extensively.
Answer: Thanks for your question. Yes, not each change in nucleotide sequence is followed by the sequence in AA sequence. So our discussion focused more on amino acid changes. For this issue, we have made revision in the manuscript.
Line 320 - Several lines above you put the statement that mutation rate in NS1 is higher, here you claimed that NS1 is conserverd. Explain it and support the statement with proper reference.
Answer: As suggested, we have added information to support the statement in the manuscript.
Line 321 - explain what does mean "Asian characteristics'?
Answer: Sorry for the confusion caused by our inadequate description, what we're trying to say here is about isolates from Asia, we have made revision in the manuscript.
Line 323-325 - General statement that not quite stem from your paper.
Answer: As suggested, we have made revision in the manuscript.
Lines 329-331 - This is a sample of what discussion should be. Mostly this section is repetition of results section. As I mention above, compare your results with other studies.
Answer: As suggested, we have made revision in the manuscript.
- Conclusion – Final statement (Lines 345-346) are not quite justified, as we know nothing about the vaccination status of the tested animals.
Answer: As suggested, we have rewritten this sentence in the manuscript.

Reviewer 2 Report
Comments and Suggestions for Authors
1. Canine parvovirus infects cats and causes disease, which is a well-known phenomenon, is an old problem, not a new discovery by the author of this article.
2. The author of this paper used domestic cats to carry out the FPV challenge experiment, which resulted in death of cats. Adequate animal ethical evaluation should be conducted before the experiment began, and the welfare of animals during the experiment should be guaranteed. I did not find any relevant description in the article and its attachment, please give a credible explanation.
3. As an article on the molecular epidemiology of FPV in China, the author only referred to articles published in English, and did not involve Chinese articles on FPV published by China researchers in Chinese journals. The views and conclusions put forward by the author are not comprehensive, and hope to make further additions.
Author Response
Response to reviewers
We would like to thank you for your helpful comments and suggestions, which were valuable to improve the quality of our manuscript. We have extensively and carefully revised the entire manuscript based on your review. A revised manuscript with the corrections highlighted using blue mark has been prepared. The reviewers’ comments are shown below followed by our pointwise responses.
Reviewer 2
- Canine parvovirus infects cats and causes disease, which is a well-known phenomenon, is an old problem, not a new discovery by the author of this article.
Answer: Thanks for your point of view and we agree with you. Canine parvovirus infects cats and causes disease, which is not a new problem. At the same time, it’s also not a new discovery in our manuscript. However, this issue has persisted in our region, with limited investigations conducted in Henan Province, China. Consequently, we embarked on research and investigation in this area to contribute data for future vaccine development and enhance prevention and control measures.
- The author of this paper used domestic cats to carry out the FPV challenge experiment, which resulted in death of cats. Adequate animal ethical evaluation should be conducted before the experiment began, and the welfare of animals during the experiment should be guaranteed. I did not find any relevant description in the article and its attachment, please give a credible explanation.
Answer: Thanks for your question. In deed, we really did not emphasize in the article, but we are approved by animal welfare center of our university. When conducting experiments involving animals, especially those that may pose risks to their welfare or result in adverse outcomes such as in the case of the FPV challenge experiment, it is essential to ensure that proper ethical considerations are addressed. Due to space constraints or the focus of the paper, the authors may have chosen not to include detailed descriptions of the ethical review process and animal welfare measures, instead prioritizing the presentation of experimental methods and results. Moreover, we have already provided the approval for publication from the patients who participated in our study is mandatory and other more detail information about animal ethics to editors.
- As an article on the molecular epidemiology of FPV in China, the author only referred to articles published in English, and did not involve Chinese articles on FPV published by China researchers in Chinese journals. The views and conclusions put forward by the author are not comprehensive, and hope to make further additions.
Answer: As suggested, we have added the references published by China researchers in Chinese journals, and discussed specific epidemic situation in China in the discussion part.

Reviewer 3 Report
Comments and Suggestions for Authors
Yu and colleagues investigated the molecular and phylogenetic analysis of parvoviruses from domestic cats during 2020 – 2022. Parvovirus infection can lead to lethal outcome in infected cats. Further, although vaccines against feline parvoviruses are available, their efficacy is considered low due to the genetic variability of different isolates. Phylogenetic analyses provide important information on the circulation of particular isolates and subgroups and might be helpful to estimate vaccine efficacy. Some studies focused on the genetic analysis of the VPS gene, whereas only limited information is given on the remaining genes of feline parvoviruses. In this study, the authors focused on the VP2 and NS1 gene, two genes involved in controlling infectivity and pathogenicity.
The manuscript is well written and the results are presented in a clear fashion.
The following minor points should be addressed to enable the publication of this study:
1. Is any information available whether the cats were vaccinated?
2. Line 177-178: Please check the percentage of nucleotide and amino acid homology given in the results as they differ from the percentage given in the discussion part (line 308), “The alignment of NS1 nucleotide sequences of the 11 isolates showed that the nucleotide homology between the 11 isolates and the reference strains ranged from 97.6% to 100%...” vs “the nucleotide and amino acid homology within and between the 21 NS1 full-length gene sequences amplified here and the 38 reference strains were 98.2% to 100% and 97.6% to 100%, respectively,”
Author Response
Response to reviewers
We would like to thank you for your helpful comments and suggestions, which were valuable to improve the quality of our manuscript. We have extensively and carefully revised the entire manuscript based on your review. A revised manuscript with the corrections highlighted using blue mark has been prepared. The reviewers’ comments are shown below followed by our pointwise responses.
Reviewer 3
- Is any information available whether the cats were vaccinated?
Answer: Thanks for your question. Before the experiment, we performed FPV specific tests on the feces and blood samples of the kittens, and the results were negative., so it can be inferred that kittens have not been vaccinated. That's why he was chosen as a test subject. Electrophoretic images of previous PCR tests are available.
- Line 177-178: Please check the percentage of nucleotide and amino acid homology given in the results as they differ from the percentage given in the discussion part (line 308), “The alignment of NS1 nucleotide sequences of the 11 isolates showed that the nucleotide homology between the 11 isolates and the reference strains ranged from 97.6% to 100%...” vs “the nucleotide and amino acid homology within and between the 21 NS1 full-length gene sequences amplified here and the 38 reference strains were 98.2% to 100% and 97.6% to 100%, respectively,”
Answer: First of all, thank you very much for your questions, and secondly, we are deeply sorry for this writing error. Thanks for your patience and guidance, we have checked the data in the manuscript and changed the content to "amplification between 21 NS1 full-length gene sequences and 38 reference strains 98.2% to 100% and 97.6% to 100%".

Reviewer 4 Report
Comments and Suggestions for Authors
This is an interesting study reporting feline parvovirus in cats in one of the Chinese provinces in recent past. After careful reading, this reviewer has following suggestions for the authors:
1. In Introduction, please provide information on FPV genome and gene functions, their significance. Why is NS1 gene important?
2. Introduction needs more information of host range and prevalence of FPV in the region. Any related studies on FPV detection in recent past in China?
3. Please provide the reference(s) if previously reported protocol(s) were used in this study.
4. Table 1: were these primers previously reported or designed by the authors?
5. Methods: more information on the phylogenetic analysis is required - substitution model used?
6. Line 154: PCR-positive for what?
7. Results: Tables 3 and 4 are not described.
8. Please provide a summarizing paragraph of results before starting the Discussion section.
9. Line 268: 'Infection rate' might not be the right term for this. I think you may say 'prevalence, based on reported data'.
10. Line 270: 'Few studies' but only one citation was provided.
11. Line 338: 'positive rate'. Improve such wordings and sentences.
12. Please improve the conclusions of the study. The current conclusions are not convincing.
13. English must be improved throughout the text.
Comments on the Quality of English LanguageEnglish language must be improved throughout the manuscript.
Author Response
Response to reviewers
We would like to thank you for your helpful comments and suggestions, which were valuable to improve the quality of our manuscript. We have extensively and carefully revised the entire manuscript based on your review. A revised manuscript with the corrections highlighted using blue mark has been prepared. The reviewers’ comments are shown below followed by our pointwise responses.
Reviewer 4
- In Introduction, please provide information on FPV genome and gene functions, their significance. Why is NS1 gene important?
Answer: Thanks for your valuable questions. According to your questions, we have supplemented the description of the manuscript.
- Introduction needs more information of host range and prevalence of FPV in the region. Any related studies on FPV detection in recent past in China?
Answer: Thank you very much for your valuable questions. We have looked up relevant information. There is not much research on China in this area at present. We also conducted this survey to provide a basis for enriching local data and disease prevention and control.
- Please provide the reference(s) if previously reported protocol(s) were used in this study.
Answer: As suggestion, we added references in the method part.
- Table 1: were these primers previously reported or designed by the authors?
Answer: Thank you very much for your valuable questions. The VP2 primers in Table 1 are designed by ourselves, and other primers are reported in other articles for reference.
- Methods: more information on the phylogenetic analysis is required - substitution model used?
Answer: Thanks to your advice, we've reconstructed the phylogenetic tree. Your advice is of great value.
- Line 154: PCR-positive for what?
Answer: Thank you for your question. PCR positive refers to the identification primer of FPV, and the samples amplified by PCR method are FPV positive. We have revised it in the manuscript, thank you again for your questions and patience.
- Results: Tables 3 and 4 are not described.
Answer: Thank you for your question. We have explained Table 3-4 in lines 231-246 of the old manuscript.
- Please provide a summarizing paragraph of results before starting the Discussion section.
Answer: Thank you for your suggestion. We have added information in the manuscript.
- Line 268: 'Infection rate' might not be the right term for this. I think you may say 'prevalence, based on reported data'.
Answer: Thank you for your suggestion. We have made some modifications in the paper to change the positive detection rate of FPV. Thank you for your patient guidance.
- Line 270: 'Few studies' but only one citation was provided.
Answer: Thank you for your question about the quotation. Because there are relatively few studies on FPV in Henan Province, we do not have much research data on this area. Thank you again for your patience and guidance.
- Line 338: 'positive rate'. Improve such wordings and sentences.
Answer: Thanks for your patient guidance, we have made modifications in the paper according to your tips.
- Please improve the conclusions of the study. The current conclusions are not convincing.
Answer: Thanks for your suggestions. We have carefully revised the conclusion. Thanks again for your suggestions and patience.
- English must be improved throughout the text.
Answer: Thanks for your advice. As suggested, we have asked a native English speaker to correct the language of the manuscript.

Round 2
Reviewer 1 Report
Comments and Suggestions for Authors
I see that paper has been corrected and improved in some aspects. Lack of internal control is some flaw of this study, however on this stage it is not possible to correct this issue. Hope that, Authors bear in mind this advice, designing future studies. Still, I have some comments. First of all, I am grateful for the detailed responses, but I would like to see the responses incorporated into the text of the manuscript.
Abstract – Line 26 – Edit this sentence to point the titer causing the CPE.
Introduction – I see that you provided quite comprehensive information concerning relation FPV-CPV-2. Even more detailed, than I expected.
Material and methods:
Line 104 - Include the reference to the supplementary figure – map.
Figure S1 – number of what ? Specify
Line 117 - ‘Yes, we verified the quantity and quality of DNA after extraction using nano-drop ultra-micro spectrophotometer. For samples with poor quality, we will extract them again.’ - Include information, how the quality and quantity of DNA were evaluated in the main text.
Line 135 – ‘means the restriction enzymes’ or sites recognized by the restriction enzymes ?
146-147 – ‘Through these computational techniques, researchers were able to elucidate the evolutionary relationships and genetic diversity within the studied viral populations’ – in my opinion this sentence is redundant.
Line 179 – again you clarified the issue that I raised last time, but you should also include this statement in main paper.
Results
Line 202 remove the symbol - | from the line
Supplementary figures 2-4 –How we can compare the similarity between your isolates in such orientated map ?
Figure 1 – Your response is not precise – I did not ask about the role of particular software in your phylogenetic analysis, but I am curious why you have included the particular sequences to the groups. You claim that these two sequences are not suitable for such grouping, answer why? What numerical values did you consider to get such conclusion?
Now, we have similar situation with the sequence - FPV/USA/2011 (JX475256.1), is it counted as member of G3 group why not G2 ? The quality of the tree need to be improved.
Line 266- black or red ?
Include the table with the viral load in manuscript or as supplementary material. Correct the notation of viral load to n x 10n copies/ug. No.1, 2 and 3, means animals or replicates ? Explain in the text, maybe better to provide the mean value +/- SE ? Did you verify if there are statistically significant differences in viral load between tested organs ?
Discussion
Lines 324-330 – Have you changed this part of the text ? Why it is not highlighted? Are there more changes in manuscript which were not pointed ?
Lines 338-345 – Still this part is repetition of the results.
Lines 346-350 – Good, but try to combine this information with your results, how they are connected ? Did you notice some mutation that may change the structure of the one of the loops ? May they have impact on the virulence ? Try to discuss your results properly.
Line 354 - Still stable mutation – in response, you wrote that this issue was corrected.
Lines 360-376 – Here is a example of acceptable discussion. I would like to see more such fragments.
Concluding, I would like to see the revised paper and rate it once more. Still there is too much doubts, to recommend this paper for publication.
Author Response
Dear reviewer,
Thank you very much for the suggestions and giving us the second chance to revise the manuscript. Responses were showed as follows:
- Abstract– Line 26 – Edit this sentence to point the titer causing the CPE.
Answer: Thanks for your suggestion, we have edited the sentence again.
- Introduction– I see that you provided quite comprehensive information concerning relation FPV-CPV-2. Even more detailed, than I expected.
Answer: Thank you for your recognition, your recognition is a great honor for us.
- Material and methods:
Line 104 - Include the reference to the supplementary figure – map.
Answer: Thank you for your reminder, we have added it in the manuscript.
Figure S1 – number of what ? Specify
Answer: Thank you for your question. The figure in S1 refers to the number of fecal samples collected, and we have revised and explained it in the figure.
Line 117 - ‘Yes, we verified the quantity and quality of DNA after extraction using nano-drop ultra-micro spectrophotometer. For samples with poor quality, we will extract them again.’ - Include information, how the quality and quantity of DNA were evaluated in the main text.
Answer: Thank you for your question. Through the nano-drop ultra-micro spectrophotometer, the specific concentration (ng/μL) of DNA could be detected, and according to the value of OD260/280 the quality of the extracted DNA could be evaluated.
Line 135 – ‘means the restriction enzymes’ or sites recognized by the restriction enzymes ?
Answer: Thank you for your question, our statement is inaccurate. Its means sites recognized by the restriction enzymes.
146-147 – ‘Through these computational techniques, researchers were able to elucidate the evolutionary relationships and genetic diversity within the studied viral populations’– in my opinion this sentence is redundant.
Answer: Thanks for your suggestion, we have deleted the sentence in the manuscript.
Line 179 – again you clarified the issue that I raised last time, but you should also include this statement in main paper.
Answer: Thanks for your suggestion, we have added the statement in the manuscript.
- Results
Line 202 remove the symbol - | from the line
Answer: Thanks for your advice, we have removed '|' in forms 1, 2 and 3.
Supplementary figures 2-4 –How we can compare the similarity between your isolates in such orientated map ?
Answer: Thank you for your question, each cell in the heatmap corresponds to a data point, and its color intensity represents the value of that data point. We think this is an intuitive way to represent large data sets. By encoding data values using color gradients, it is easier to identify patterns, trends, and outliers at a glance.
Figure 1 – Your response is not precise – I did not ask about the role of particular software in your phylogenetic analysis, but I am curious why you have included the particular sequences to the groups. You claim that these two sequences are not suitable for such grouping, answer why? What numerical values did you consider to get such conclusion?
Answer: First of all, thank you very much for your question, and we are sorry that we have ignored the scores of these two sequences. According to the scores, they should be in group G1, and we will modify them in the figure. In addition, we chose Bootstrap consensus Tree before, and then chose original tree as the format. We did not change other values, but only changed the presentation form of the tree. Thank you very much for your question. Let us also be aware of the different forms of evolution tree presentation and different perspectives of analysis. This question deserves our deep thinking. Thank you again for your patience and questions.
Now, we have similar situation with the sequence - FPV/USA/2011 (JX475256.1), is it counted as member of G3 group why not G2? The quality of the tree need to be improved.
Answer: Yes, the question you raised is very good. We ignored the value and did not check it carefully. This problem is the same as the above problem. Your question This is very worthy of our deep thinking, thank you again for your patience and questions.
Line 266- black or red ?
Answer: Thank you for your question. This is our mistake. We have marked in black in the figure, and we have modified it in the figure. Thanks again for your help.
Include the table with the viral load in manuscript or as supplementary material. Correct the notation of viral load to n x 10n copies/ug. No.1, 2 and 3, means animals or replicates ? Explain in the text, maybe better to provide the mean value +/- SE ? Did you verify if there are statistically significant differences in viral load between tested organs ?
Answer: Thank you for your suggestion. The table with the viral load as supplementary material (Table S4) was included. No.1, 2 and 3, means animals or replicates. As suggested, we provided the mean and SE value. We didn’t verify if there are statistically significant differences in viral load between tested organs.
- Discussion
Lines 324-330 – Have you changed this part of the text ? Why it is not highlighted? Are there more changes in manuscript which were not pointed ?
Answer: Yes, we have checked the content and did modify it, but we forgot to highlight the mark because of our carelessness. We will carefully check the modification. Thank you again for your patience and guidance.
Lines 338-345 – Still this part is repetition of the results.
Answer: Thanks for your suggestion, we have deleted and modified this part in the manuscript.
Lines 346-350 – Good, but try to combine this information with your results, how they are connected ? Did you notice some mutation that may change the structure of the one of the loops ? May they have impact on the virulence ? Try to discuss your results properly.
Answer: Thanks for your suggestion, we have discussed it in the previous revision and made some modifications based on the original manuscript.
Line 354- Still stable mutation – in response, you wrote that this issue was corrected.
Answer: Thank you for your suggestion. We have revised in the manuscript.
Lines 360-376 – Here is a example of acceptable discussion. I would like to see more such fragments.
Answer: Thank you for your recognition, we are very honored to get your recognition, we will carefully revise our paper according to your guidance, your professionalism deserves our admiration. And thank you for the opportunity you've given us. Thank you so much again.

Reviewer 2 Report
Comments and Suggestions for Authors
I am very glad that the authors have given adequate answers to all of my questions. I have no new comments and agree to publish this article.
Author Response
Thank you for your advice and final approval.
Reviewer 4 Report
Comments and Suggestions for Authors
Thanks for the revision and considering previous suggestions. Well done!
Author Response

(The authors gave the same response as above.)

Round 3
Reviewer 1 Report
Comments and Suggestions for Authors
Just few issues.
First of all – Supplementary figures. You have not provided the supplementary figure with map (I suppose the Fig S1.). Instead you put the reference to figure S2, which is heatmap. Revise the order of the figures. I suppose that you upload the suypplementary materials from the first version of paper, as there still is 'year of separation' instead of isolation. Correct it.
Second – Thank you for the details about quantity and quality measurement. Please include these information in the manuscript after description of DNA extraction.
We still have problem with the phylogenetic tree. Sequence - FPV/USA/2011 (JX475256.1) do you think that this sequence should be included in the G3 group ? Could you provide better separation between branches ?
Line 354-35 I am not quite understand - you got complete full sequence of these genes, so why do hypothesize that the viral DNA may be incomplete ?
Concluding, In my opinion after correction of issues mentioned above, the paper should meet the requirements of publication. Nevertheless, I would like to see the revised paper before publication
Author Response
For research article
|
Response to Reviewer 1 Comments
|
||
|
Summary |
|
|
|
Thank you very much for taking the time to review this manuscript. Please find the detailed responses below and the corresponding revisions highlighted in the re-submitted files.
|
||
|
Point-by-point response to Comments and Suggestions for Authors |
||
|
Comments 1: First of all – Supplementary figures. You have not provided the supplementary figure with map (I suppose the Fig S1.). Instead you put the reference to figure S2, which is heatmap. Revise the order of the figures. I suppose that you upload the suypplementary materials from the first version of paper, as there still is 'year of separation' instead of isolation. Correct it. |
||
|
Response 1: Thank you for pointing this out. Therefore, We have corrected the errors you mentioned. |
||
|
Comments 2: Second – Thank you for the details about quantity and quality measurement. Please include these information in the manuscript after description of DNA extraction. |
||
|
Response 2: Agree. We have, accordingly, modified the information in the manuscript. |
||
|
Comments 3: We still have problem with the phylogenetic tree. Sequence - FPV/USA/2011 (JX475256.1) do you think that this sequence should be included in the G3 group ? Could you provide better separation between branches ? |
||
|
Response 3: Thank you for pointing this out. Sequence FPV/USA/2011 (JX475256.1) should not be included in the G3 group, due to typographical problems caused by the background color misalignment, which we have modified in the manuscript, please refer to it. |
||
|
Comments 4: Line 354-35 I am not quite understand - you got complete full sequence of these genes, so why do hypothesize that the viral DNA may be incomplete ? |
||
|
Response 4: Agree. We have, accordingly, modified the information in the manuscript. |
||
|
Comments 5: Concluding, In my opinion after correction of issues mentioned above, the paper should meet the requirements of publication. Nevertheless, I would like to see the revised paper before publication. |
||
|
Response 5: Thank you for your help and support in this manuscript, the quality of our manuscript has been greatly improved by your suggestion. For your rigorous academic attitude and profound scientific research skills to offer our deep respect. Thank you again for your help and support. |
||

Round 4
Reviewer 1 Report
Comments and Suggestions for Authors
Line 136-139 - could be or were ?
Remove previous version of phylogenetic trees.
Author Response
Comments 1: Line 136-139 - could be or were ?
Response 1: Thank you for pointing this out. It should be ‘were ’. We have corrected the error in the updated manuscript.
Comments 2: Remove previous version of phylogenetic trees.
Response 2: Thank you for pointing this out. We have replaced the old phylogenetic tree with an updated one.